# Calcium is an essential cofactor for metal efflux by the ferroportin transporter family

Chandrika N. Deshpande[1], T. Alex Ruwe [2,3], Ali Shawki[2,3,6], Vicky Xin[1], Kyle R. Vieth[2], Erika V. Valore[4], Bo Qiao[4], Tomas Ganz [4,5], Elizabeta Nemeth[4], Bryan Mackenzie [2,3] & Mika Jormakka [1]

Ferroportin (Fpn)—the only known cellular iron exporter—transports dietary and recycled iron into the blood plasma, and transfers iron across the placenta. Despite its central role in iron metabolism, our molecular understanding of Fpn-mediated iron efflux remains incomplete. Here, we report that $Ca^{2+}$ is required for human Fpn transport activity. Whereas iron efflux is stimulated by extracellular $Ca^{2+}$ in the physiological range, $Ca^{2+}$ is not transported. We determine the crystal structure of a $Ca^{2+}$-bound BbFpn, a prokaryotic orthologue, and find that $Ca^{2+}$ is a cofactor that facilitates a conformational change critical to the transport cycle. We also identify a substrate pocket accommodating a divalent transition metal complexed with a chelator. These findings support a model of iron export by Fpn and suggest a link between plasma calcium and iron homeostasis.

---

[1] Structural Biology Program, Centenary Institute, Sydney Medical School, University of Sydney, Sydney, NSW 2042, Australia. [2] Department of Pharmacology & Systems Physiology, University of Cincinnati College of Medicine, Cincinnati 45267 OH, USA. [3] Systems Biology & Physiology Program, University of Cincinnati College of Medicine, Cincinnati 45267 OH, USA. [4] Department of Medicine, David Geffen School of Medicine at University of California, Los Angeles 90095 CA, USA. [5] Department of Pathology, David Geffen School of Medicine at University of California, Los Angeles 90095 CA, USA. [6] Present address: Division of Biomedical Sciences, University of California-Riverside School of Medicine, 900 University Avenue, Riverside, CA 92521, USA. These authors contributed equally: Chandrika N. Deshpande, T. Alex Ruwe. Correspondence and requests for materials should be addressed to B.M. (email: bryan.mackenzie@uc.edu) or to M.J. (email: m.jormakka@centenary.org.au)

Ferroportin (Fpn) activity plays a central role in human iron homeostasis on both systemic and cellular levels[1,2]. In vertebrates, Fpn is subject to post-translational regulation by the hormone hepcidin, a key event in iron pathophysiology. Hepcidin binds to active Fpn and signals the endocytosis and proteolysis of the Fpn–hepcidin complex when systemic iron levels are high[3]. Fpn gene knockout is embryonically lethal in mice[4] and human mutations that affect Fpn activity or sensitivity to hepcidin, or dysregulation of the Fpn–hepcidin interplay, produce iron disorders, including iron-restricted anaemias and hemochromatosis[5].

Although the homeostatic role of Fpn has been extensively characterized, lack of structural and functional information has impeded our detailed understanding, and thus the pharmacological targeting, of this system (reviewed in refs.[6,7]). In a significant advancement, recent crystal structures of a bacterial Fpn ortholog (Bdellovibrio bacteriovorous; BbFpn) provided clues to the hepcidin-binding mode in vertebrates and confirmed a major facilitator superfamily (MFS) type fold, comprising 12 transmembrane (TM) helices organized into N-terminal (TM1–6) and C-terminal (TM7–12) domains[8]. In most MFS transporters, the substrate-binding site is formed at the interface of the two domains and is alternately accessible from either side of the membrane[9]. The transport cycle is viewed as a series of ligand-induced conformational changes that include open outward and open inward states. Both states were structurally determined for BbFpn, revealing a putative substrate-binding site deep in the N-terminal domain[8]; however, the binding site assignment was provisional, in part due to its unorthodox location. Furthermore, ion coupling and the molecular mechanism underlying alternating-access transport remain unknown.

Here, we use biophysical analyses, functional assays, and site-directed mutagenesis to explore ion coupling and to identify the substrate site in Fpn. We demonstrate that removal of extracellular $Ca^{2+}$ abolishes Fpn-mediated iron efflux in Xenopus oocyte and human (HEK) expression systems. Detailing the nature of the $Ca^{2+}$ dependence, we present the crystal structure of a $Ca^{2+}$-bound BbFpn protein, which supports a model in which $Ca^{2+}$ is a required cofactor that facilitates a conformational change critical to the transport cycle. The structure of the $Ca^{2+}$-bound BbFpn also reveals a putative metal-binding pocket, and mutagenesis of the corresponding site in the human protein changes the metal specificity of Fpn-mediated transport.

## Results

**Calcium activates Fpn-mediated iron efflux.** We considered the hypothesis that Fpn functions as an iron/cation antiporter. Metal transport by both BbFpn and human Fpn was previously shown to be pH-sensitive, but independent of the TM $H^+$ gradient[8,10]. In addition, BbFpn-mediated metal transport was $Na^+$-independent[8]. Here we also demonstrate that human Fpn-mediated $^{55}Fe$ efflux in RNA-injected Xenopus oocytes is unaffected by $Na^+$ replacement with choline (Fig. 1a). Data for BbFpn and human Fpn therefore do not support a role for monovalent cations in driving metal transport.

We next removed or replaced extracellular divalent cations and found that Fpn-mediated efflux of $^{55}Fe$ from oocytes was dependent on extracellular $Ca^{2+}$ (Fig. 1b). Extracellular $Ca^{2+}$ stimulated $^{55}Fe$ efflux with half-maximal $Ca^{2+}$ concentration ($K_{0.5}^{Ca}$) of 0.8 ± (SEM) 0.2 mM (Fig. 1c). We also examined the efflux of Co and Zn, both of which are Fpn substrates[10], and

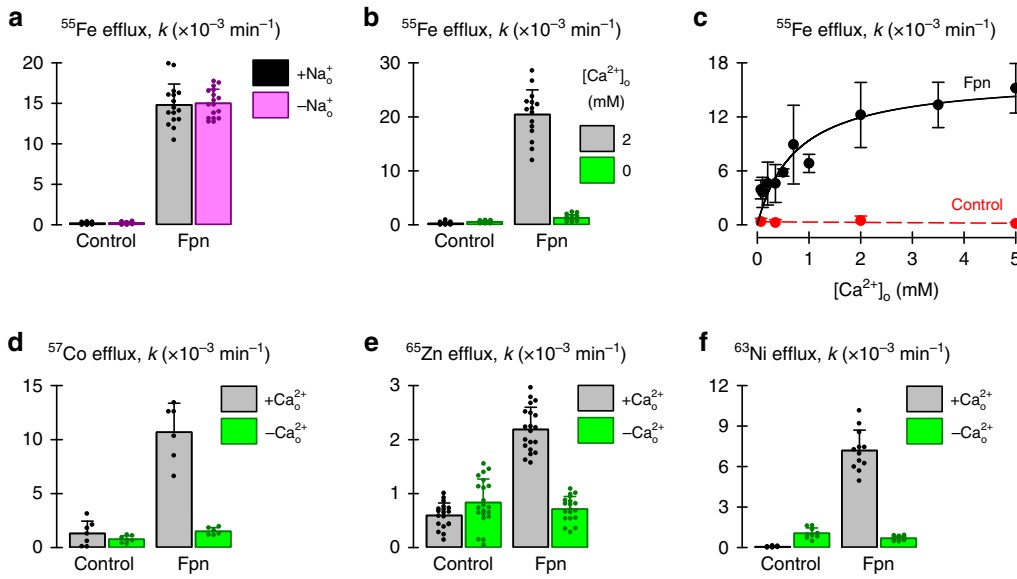

**Fig. 1** Calcium activates Fpn-mediated iron efflux in Xenopus oocytes. **a** First-order rate constants ($k$) for $^{55}Fe$ efflux from control oocytes and oocytes expressing human Fpn, in the presence (+$Na^+_o$) or absence (−$Na^+_o$) of 100 mM extracellular $Na^+$; $n = 10, 14, 16, 17$ oocytes per group (left to right). Two-way ANOVA revealed no interaction ($P = 0.86$); within Fpn, +$Na^+_o$ did not differ from −$Na^+_o$ ($P = 0.73$). **b** $^{55}Fe$ efflux from control oocytes and oocytes expressing human Fpn in the presence of 2 mM extracellular $Ca^{2+}$ or in its absence (i.e. 0 mM $Ca^{2+}$ plus 1 mM EGTA) ($n = 16, 16, 15, 16$). Two-way ANOVA: interaction ($P < 0.001$). **c** $^{55}Fe$ efflux from control oocytes (red) and oocytes expressing Fpn (black) as a function of $[Ca^{2+}]_o$ (control, $n = 30$; Fpn, $n = 84$; i.e. 6–9 oocytes per group at each $[Ca^{2+}]_o$). Fpn data were fit by Eq. (2): $K_{0.5}^{Ca} = 0.8 ± (SEM)$ 0.2 mM, $V_{max}$ occurred at $k = (16 ± 2) × 10^{-3}$ min$^{-1}$ ($r^2 = 0.88$, $P < 0.001$). **d** $^{57}Co$ efflux from control oocytes and oocytes expressing Fpn in the presence of 2 mM extracellular $Ca^{2+}$ or in its absence (i.e. 0 mM $Ca^{2+}$ plus 1 mM EGTA) ($n = 7, 7, 6, 6$). Two-way ANOVA: interaction ($P < 0.001$). **e** $^{65}Zn$ efflux from control oocytes and oocytes expressing Fpn injected with 50 nl of 50 μM $^{65}Zn$ in the presence of 2 mM extracellular $Ca^{2+}$ or in its absence (i.e. 0 mM $Ca^{2+}$ plus 1 mM EGTA) ($n = 19, 21, 20, 19$). Two-way ANOVA: interaction ($P < 0.001$). **f** $^{63}Ni$ efflux from control oocytes and oocytes expressing Fpn in the presence of 2 mM extracellular $Ca^{2+}$ or in its absence (i.e. 0 mM $Ca^{2+}$ plus 1 mM EGTA) ($n = 10, 9, 12, 12$). Two-way ANOVA: interaction ($P < 0.001$). All bar graphs indicate mean, SD, and individual scores; data in (**c**) are mean, SD

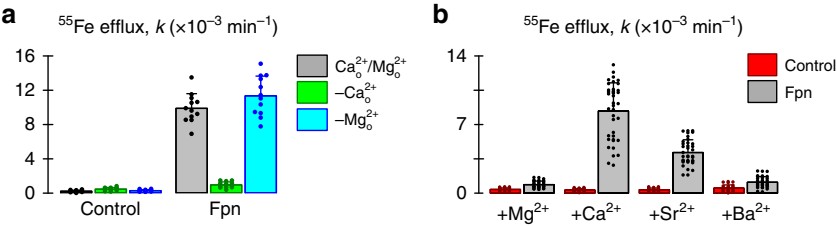

**Fig. 2** Alkaline-earth metal selectivity. **a** $^{55}Fe$ efflux from control oocytes and oocytes expressing Fpn, in the presence of extracellular calcium (2 mM) and magnesium (1 mM), the absence of extracellular calcium ($-Ca^{2+}_o$ i.e. 0 mM $Ca^{2+}$ plus 1 mM EGTA), or the absence of extracellular magnesium ($-Mg^{2+}_o$) i.e. 0 mM $Mg^{2+}$ plus 1 mM EDTA) ($n = 12$ in each group). Two-way ANOVA: interaction ($P < 0.001$); within Fpn, all groups differ from one another ($P \leq 0.004$). **b** $^{55}Fe$ efflux from control oocytes and oocytes expressing Fpn in the presence of 3 mM extracellular magnesium ($+Mg^{2+}$), or 1 mM $Mg^{2+}$ plus 2 mM calcium ($+Ca^{2+}$), strontium ($+Sr^{2+}$), or barium ($+Ba^{2+}$) ($n = 30, 39, 28, 32, 27, 36, 25, 32$). Two-way ANOVA: interaction ($P < 0.001$). In multiple pairwise comparisons vs. control: Fpn differed from control in the case of $Ca^{2+}$ and $Sr^{2+}$ ($P < 0.001$), but not $Mg^{2+}$ ($P = 0.14$) or $Ba^{2+}$ ($P = 0.082$); within Fpn, all conditions differed from '$+Ca^{2+}$' ($P < 0.001$). Bar graphs indicate mean, SD, and individual scores

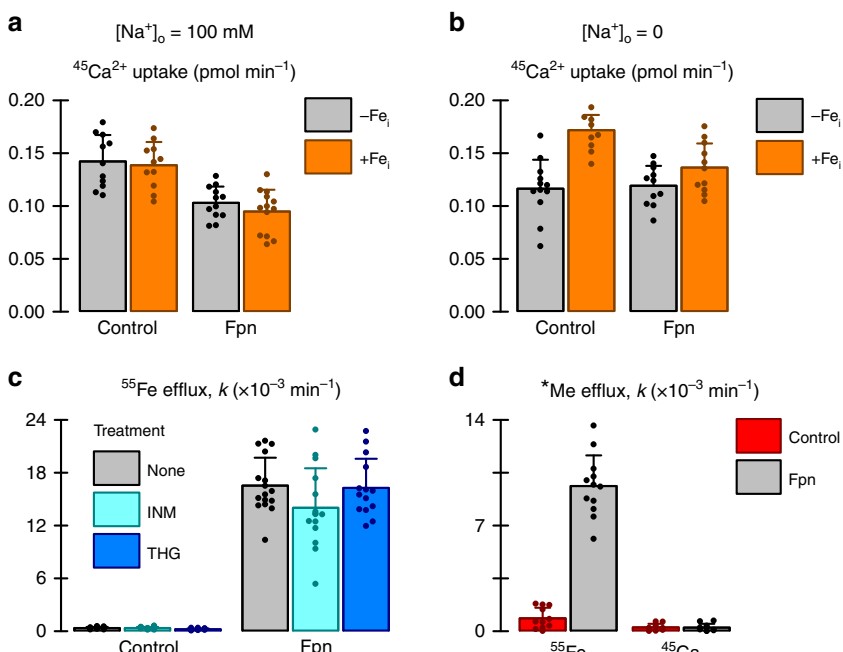

**Fig. 3** No evidence to support that calcium is a transported substrate of Fpn. **a** Uptake of 100 μM $^{45}Ca^{2+}$ in control oocytes and oocytes expressing Fpn that were injected with either vehicle alone ($-Fe_i$) or 50 nl of 50 μM Fe ($+Fe_i$) ($n = 12, 12, 13, 14$, left to right). Two-way ANOVA: no interaction ($P = 0.70$). **b** Uptake of 100 μM $^{45}Ca^{2+}$ in control oocytes and oocytes expressing Fpn in the absence of extracellular $Na^+$ ($Na^+$ was replaced by choline) ($n = 12, 8, 11, 12$). Oocytes were injected with either vehicle alone ($-Fe_i$) or 50 nl of 50 μM Fe ($+Fe_i$). Two-way ANOVA: interaction ($P = 0.008$); within Fpn, $+Fe_i$ did not differ from $-Fe_i$ ($P = 0.069$). **c** Effect of 1 μM ionomycin (IMN) or 2 μM thapsigargin (THG) on $^{55}Fe$ efflux ($n = 15, 14, 14, 16, 15, 14$). All media contained 0.2% DMSO. Two-way ANOVA: no interaction ($P = 0.13$). **d** Efflux of $^{55}Fe$ or $^{45}Ca$ efflux from control oocytes and oocytes expressing Fpn ($n = 11, 12, 9, 10$). Two-way ANOVA: interaction ($P < 0.001$); within $^{45}Ca$, Fpn did not differ from control ($P = 0.98$). All bar graphs indicate mean, SD, and individual scores

found that extracellular $Ca^{2+}$ also activated Fpn-mediated $^{57}Co$ and $^{65}Zn$ efflux (Fig. 1d, e). We have recently found that human Fpn also mediates Ni transport. We provide here evidence of Fpn-mediated $^{63}Ni$ efflux and show that it too is activated by extracellular $Ca^{2+}$ (Fig. 1f).

Since we had used EGTA to chelate extracellular $Ca^{2+}$ (Fig. 1b, d–f), we considered whether the observed inhibition of metal efflux was due to chelation of $Mg^{2+}$ by EGTA; however, specific chelation of $Mg^{2+}$ by EDTA in the presence of $Ca^{2+}$ had no effect on Fpn-mediated $^{55}Fe$ efflux (Fig. 2a). To further characterize the cation specificity, we measured $^{55}Fe$ efflux in the presence of other alkaline-earth metals (in $Ca^{2+}$-free media) and found that strontium could partially substitute for $Ca^{2+}$ but that magnesium and barium could not (Fig. 2b). This is reminiscent of many other

$Ca^{2+}$-binding proteins, which commonly have the capacity to bind to $Sr^{2+}$ due to their similar chemical properties[11,12]. Selectivity against other ions is achieved by the chemistry between the ion and binding site, e.g. nature of coordinating residues, bond lengths, coordination geometry, and level of ion hydration (reviewed in ref.[13]).

The ionized $Ca^{2+}$ concentration in blood plasma (reference range 1.0–1.2 mM)[14] is ~$10^4$-fold higher than typical intracellular $Ca^{2+}$ concentrations (100–200 nM), presenting a large electrochemical potential difference, and thus making $Fe^{2+}/Ca^{2+}$ antiport an attractive transport model by which Fpn could operate. However, we found that neither Fpn expression nor intracellular iron injection stimulated $^{45}Ca^{2+}$ uptake in oocytes (Fig. 3a). We also tested $^{45}Ca^{2+}$ uptake in the

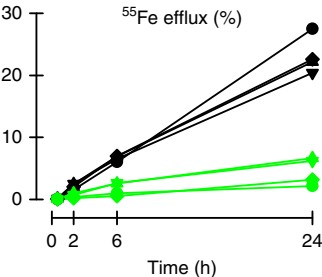

**Fig. 4** Calcium activates Fpn-mediated iron efflux in a HEK cell expression system. HEK293T cells expressing Fpn were loaded with $^{55}$Fe, and $^{55}$Fe efflux measured in media containing 1 mM $Ca^{2+}$ (black) or 0 $Ca^{2+}$ +1 mM EGTA (green). Data, normalized by $^{55}$Fe content at time = 0, are displayed as induced minus uninduced in the same preparation. Each symbol-and-line plot represents one of four independent preparations. Two-way RM ANOVA: interaction of $[Ca^{2+}]_o$ and time ($P < 0.001$); $Ca^{2+}$ significantly activated $^{55}$Fe efflux over 6 h ($P = 0.002$) and 24 h ($P < 0.001$), but not at earlier time points ($P \geq 0.24$)

**Table 1 Data collection and refinement statistics (molecular replacement)**

|  | BbFpn $Ca^{2+}$-bound (PDBID: 4BTX) |
|---|---|
| *Data collection* | |
| Space group | $P2_1$ |
| *Cell dimensions* | |
| $a, b, c$ (Å) | 57.07, 54.11, 72.44 |
| $\alpha, \beta, \gamma$ (°) | 90, 103.6, 90 |
| Resolution (Å) | 42.9-3.20 (3.31-3.20)[a] |
| $R_{meas}$ | 0.240 (0.831) |
| $I / \sigma I$ | 6.29 (1.78) |
| Completeness (%) | 96.2 (98.0) |
| Redundancy | 3.3 (3.2) |
| *Refinement* | |
| Resolution (Å) | 42.4-3.20 |
| No. of reflections | 6986 (700) |
| $R_{work}/R_{free}$ | 22.3/25.5 |
| No. of atoms | |
| Protein | 2810 |
| Ligand/ion | 45 |
| Water | 7 |
| *B*-factors | |
| Protein | 41.0 |
| Ligand/ion | 40.3 |
| Water | 36.1 |
| R.m.s. deviations | |
| Bond lengths (Å) | 0.002 |
| Bond angles (°) | 0.53 |

A single crystal was used for the structure. The average *B* factor was calculated for all non-hydrogen atoms. r.m.s.d. of bond is the root-mean-square deviation of the bond angle and length
[a]Values in parentheses are statistics of the highest-resolution shell

absence of extracellular $Na^+$ in order to exclude any contribution of the endogenous $Na^+/Ca^{2+}$ exchanger, and again observed no Fpn-stimulated $^{45}Ca^{2+}$ uptake (Fig. 3b). Pharmacologically raising the cytosolic $Ca^{2+}$ concentration as follows had no effect on $^{55}$Fe efflux (Fig. 3c): we used ionomycin, a $Ca^{2+}$ ionophore, and thapsigargin, an inhibitor of the endoplasmic reticulum $Ca^{2+}$ ATPase, to raise the cytosolic $Ca^{2+}$ concentration and consequently decrease the outward-to-inward $Ca^{2+}$ electrochemical potential gradient. If Fpn were a $Fe^{2+}/Ca^{2+}$

antiporter, this maneuver should inhibit $^{55}$Fe efflux, but it did not. In addition, Fpn expression in oocytes did not stimulate $^{45}$Ca efflux (Fig. 3d) whereas some $^{45}$Ca efflux would be expected if the $Ca^{2+}$-binding site were exposed at the intracellular face.

We confirmed the $Ca^{2+}$-dependence of Fpn activity in a mammalian expression system. Removal of extracellular $Ca^{2+}$ abolished Fpn-mediated $^{55}$Fe efflux in HEK cells stably transfected with a tet-inducible human Fpn construct (Fig. 4). Therefore, our data obtained from two heterologous expression systems demonstrate that $Ca^{2+}$ is a required activator of human Fpn but that $Ca^{2+}$ is not transported.

**Crystal structure of a $Ca^{2+}$-bound BbFpn**. To delineate the functional rationale for the $Ca^{2+}$ dependence of Fpn, we obtained a new crystal form of BbFpn in the presence of $CaCl_2$ and determined the structure at a resolution of 3.2 Å (Table 1; Supplementary Fig. 1). The asymmetric unit consists of a single monomer comprised of 12 TM helices organized into two domains, as is typical of MFS transporters (Fig. 5a, b)[15,16]. The structure is closed on the extracellular side and open to the cytoplasmic side, thus representing an open inward conformation.

The structure revealed an ion-binding site localized in the center of the N-terminal domain. We tentatively ascribed this site to $Ca^{2+}$ based on the constituents of the crystallization condition (see Methods), and hereafter refer to the structure as $Ca^{2+}$-bound. The $Ca^{2+}$-binding site is discontinuous and formed by residues localized on TM1 (D24), TM3 (Q84), and TM6 (N196, E203), all of which are strictly conserved within the Fpn transporter family (Fig. 5a, b; Supplementary Figs. 2, 3). All of the side-chain oxygen atoms, except for that of E203, are arranged on roughly the same plane along the membrane normal, whereas E203 forms an apparent bidentate coordination with $Ca^{2+}$ from the intracellular side. At the present resolution, the overall coordination sphere resembles that of a tetrahedral pyramid; however, it is possible that there could be unresolved water molecules that complete the coordination sphere, as in other structures[17,18].

The previous structures of BbFpn were crystallized at pH 8 and pH 5 for the outward and inward facing structures, respectively. The low pH at which the inward facing structure was crystallized may unnaturally promote that state by having weakened salt bridges and hydrogen bonds that would otherwise stabilize the outward facing state[8]. In the structure presented here, crystallized at pH 8, the presence of $Ca^{2+}$ appears to be catalyzing the conformational change to the inward conformation. Although the $Ca^{2+}$-bound structure aligns well with the previous inward facing structure (PDB entry 5AYO; root-mean-square distance of 1.33 Å over 362 Cα atoms), there are significant differences in and around the $Ca^{2+}$ site. Most prominently, the coordination of $Ca^{2+}$ by N196 and E203 produces a distinct kink in the TM6 helix between these two residues, resulting in a marked displacement of E203—the carboxylate group is shifted by ~5 Å (Fig. 5c, d). We also observed that TM1 and TM4 tilt an additional ~15° towards the $Ca^{2+}$-binding site when $Ca^{2+}$ is present (Supplementary Fig. 4), thereby fully occluding the $Ca^{2+}$-binding site.

To confirm $Ca^{2+}$ binding, we used isothermal titration calorimetry (ITC) and purified protein. We measured high affinity $Ca^{2+}$ binding to wild type BbFpn ($K_d^{Ca} = 12 \pm 0.4$ μM; Fig. 5e and Supplementary Fig. 5a), whereas under the same experimental conditions, there was no observable binding of the alkali metal $K^+$ or alkaline-earth metal $Mg^{2+}$ (Supplementary Fig. 5b, c). $Sr^{2+}$ was bound by BbFpn at lower affinity than that

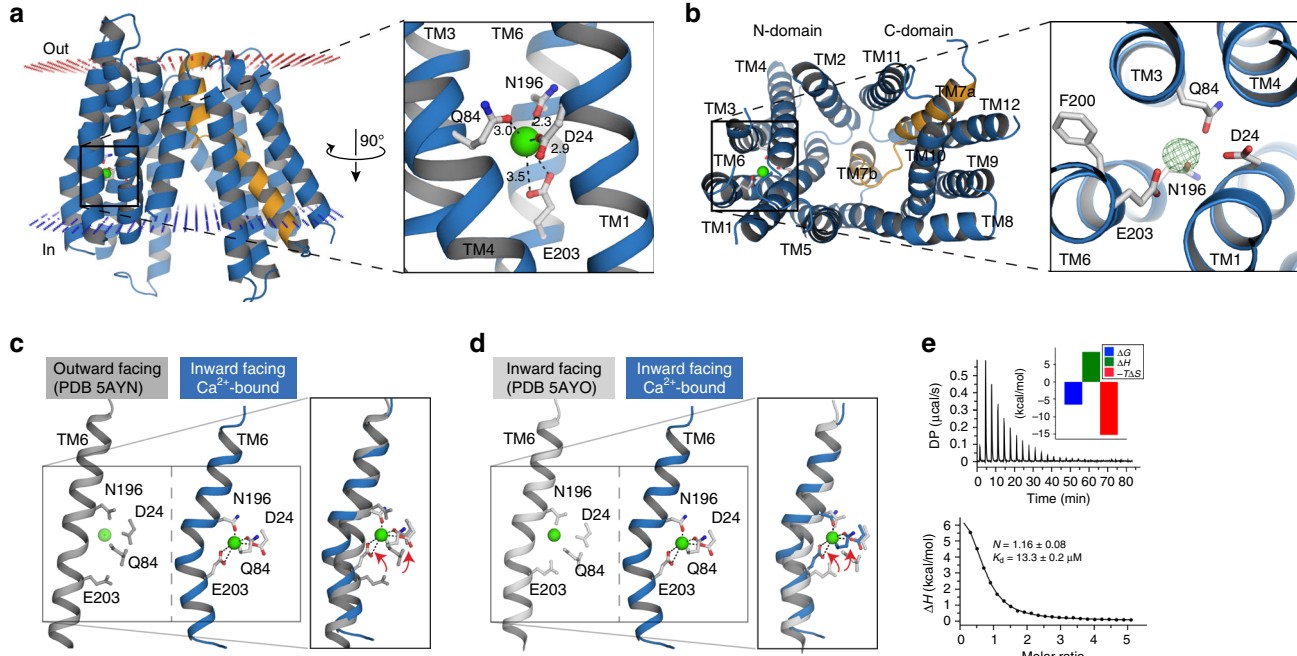

**Fig. 5** Crystal structure of a $Ca^{2+}$-bound BbFpn reveals a conserved binding site. **a** Overall structure of the $Ca^{2+}$-bound inward facing conformation of BbFpn and close up view of the $Ca^{2+}$ site (right). Approximate location of the membrane bilayer is illustrated with red (outside) and blue (inside) dots, generated using the PPM server[41]. TM7 is highlighted in orange. The $Ca^{2+}$ ion is colored in green, and coordinating residues are shown in ball-and-stick. Some of the distances are longer than the reported ideal $Ca^{2+}$–O distance (~2.5 Å)[42], although this is not unprecedented considering the resolution of the structure[17]. **b** Intracellular view (left) and $F_o$–$F_c$ omit map of the Ca site (right; contoured at 5.5σ). **c, d** Binding of the $Ca^{2+}$ ion leads to a kink forming in TM6, likely promoted by the $Ca^{2+}$ coordinating N196 and E203. **e** Representative ITC experiment between wild type BbFpn/$Ca^{2+}$. Stoichiometry (N) and dissociation constant ($K_d$) of each interaction are indicated. The variations in free energy ($\Delta G$), enthalpy ($\Delta H$), and entropy ($-T\Delta S$) are respectively, in blue, green, and red in the histogram

for $Ca^{2+}$ ($K_d^{Sr} = 46 \pm 3.3$ μM; Supplementary Fig. 5d), in agreement with what we observed in our oocyte model. To further characterize the binding site, we generated individual alanine mutants of the putative $Ca^{2+}$ coordinating residues in BbFpn and assessed them by ITC. These mutations—D24A, N196A, and E203A—resulted in the complete loss of both $Ca^{2+}$ and $Sr^{2+}$ binding (Supplementary Fig. 5e–j). The mutants Q84A and Q84N could not be purified in sufficient quantities for ITC; however, we observed no $Ca^{2+}$ or $Sr^{2+}$ binding by a Q84E mutant (Supplementary Fig. 5k, l).

The $Ca^{2+}$ site partially overlaps with the previously assigned substrate site (Supplementary Fig. 6)[8]. The latter site, comprising T20, D24, N196, S199, and F200, was assigned based on crystal soaking with Fe (5 mM) and the presence of a $K^+$ ion in the absence of metal soaking ([$K^+$] = 200–300 mM in the crystallization condition; Supplementary Fig. 6). Based on observations presented here, however, we propose that the previously assigned site non-specifically binds cations only at extremely high concentrations and is therefore not of physiological interest.

**Mutagenesis of putative $Ca^{2+}$-coordinating residues in Fpn**. By using ITC with a thermostabilized mouse Fpn protein construct (Fpn-C2; Deshpande et al. 2018), we confirmed $Ca^{2+}$ binding also in the mammalian orthologue ($K_d^{Ca} = 2.7 \pm 0.7$ μM; Supplementary Fig. 7a), whilst we observed no binding to purified Fpn-C2-E219A mutant protein (Supplementary Fig. 7b). We also examined the functional impact of mutating the orthologous $Ca^{2+}$-coordinating residues in human Fpn (i.e. residues D39, Q99, N212, and E219; Supplementary Fig. 2). Mutagenesis of any one of the putative $Ca^{2+}$ coordinating residues either (i) abolished iron-transport activity (D39E, E219D), or (ii) resulted in modest

residual activity (Q99A, N212Q) that was enhanced by increasing the extracellular $Ca^{2+}$ concentration ten-fold (Fig. 6a). By visualizing GFP fluorescence, we determined that all mutants were expressed at the oocyte perimeter at roughly the same level as wildtype Fpn (Supplementary Fig. 8a, b). Q99A exhibited decreased apparent affinity for extracellular $Ca^{2+}$ ($K_{0.5}^{Ca} = 2.4$ mM) compared with wildtype Fpn ($K_{0.5}^{Ca} = 0.9$ mM) (Fig. 6b). Transposing the glutamic acid residue from 219 to other residues within the $Ca^{2+}$-binding site (i.e. D39E/E219D and Q99E/E219D double mutants), thus displacing the putative coordination sphere, failed to rescue transport activity abolished by the E219D mutation (Fig. 6c). These results are consistent with a critical role for D39 and E219 in Fpn activity and provide compelling evidence that Q99 and N212 participate in $Ca^{2+}$ binding. Therefore, the results of crystallographic studies, ITC measurements, and results of transport assays are in agreement and together support the conclusion that $Ca^{2+}$ activates metal efflux by the Fpn protein family.

**Proposed substrate binding site in the C-terminal domain**. Our observation that the $Ca^{2+}$-binding site occupies the N-terminal domain led us to reexamine the location of the substrate-binding site. Upon examination of the $Ca^{2+}$-bound BbFpn structure, we identified a proposed substrate-binding site in the C-terminal domain. A distinguishing feature of this domain is an unwound segment in the middle of TM7 dividing the helix into TM7a and TM7b (Fig. 5a, b). Together with residues from the surrounding helices (TM8, TM10, and TM11), the unwound segment forms a large pocket (Fig. 7a) situated in an area in which the substrate-binding site has been defined in other MFS transporters[19]. In addition, unwound segments of TM helices are a common structural feature of substrate-binding sites in a diverse range of

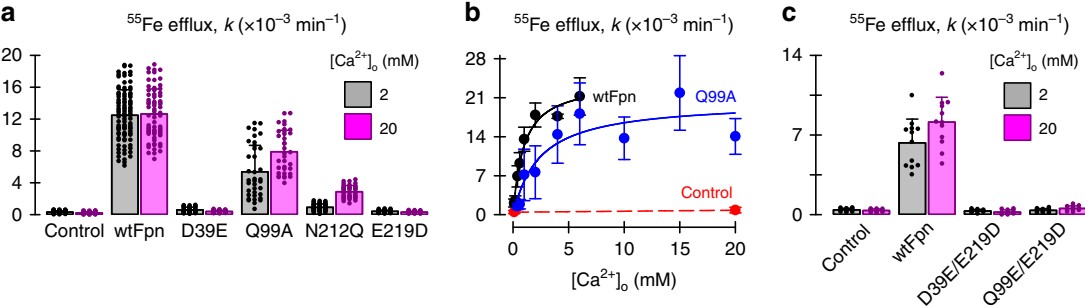

**Fig. 6** Mutagenesis of putative $Ca^{2+}$-coordinating residues in human Fpn. **a** $^{55}Fe$ efflux from control oocytes and oocytes expressing wildtype or mutant Fpn in media containing 2 or 20 mM $Ca^{2+}$ ($n = 77, 54, 96, 69, 37, 35, 37, 35, 56, 39, 34, 29$, left to right). Two-way ANOVA: interaction, $P < 0.001$; Q99A and N212Q differed from control ($P < 0.001$) but D39E and E219D did not differ from control ($P \geq 0.78$). Raising $[Ca^{2+}]_o$ from 2 to 20 mM increased $^{55}Fe$ efflux for Q99A and N212Q ($P \leq 0.001$) but not for any other mutant ($P \geq 0.61$). Relative expression levels were estimated from GFP fluorescence (Supplementary Fig. 7A). **b** $^{55}Fe$ efflux from control oocytes (red, $n = 22$) and oocytes expressing wildtype Fpn (black, $n = 62$) or Q99A (blue, $n = 76$) as a function of $[Ca^{2+}]_o$ (i.e. 6–9 oocytes at each concentration). Data were fit by Eq. (2): wildtype Fpn, $K_{0.5}^{Ca} = 0.9 \pm$ (SEM) 0.1 mM, $V_{max}$ occurred at $k = (24 \pm 1) \times 10^{-3}$ min$^{-1}$ ($r^2 = 0.97$, $P < 0.001$); and Q99A, $K_{0.5}^{Ca} = 2.4 \pm 0.4$ mM, $V_{max}$ occurred at $k = (20 \pm 1) \times 10^{-3}$ min$^{-1}$ ($r^2 = 0.83$, $P < 0.001$). **c** $^{55}Fe$ efflux from control oocytes and oocytes expressing wildtype or mutant Fpn ($n = 12, 11, 12, 12, 11, 12, 12, 10$). Two-way ANOVA: interaction, $P = 0.009$; neither mutant differed from control at either 2 or 20 mM $Ca^{2+}$ ($P \geq 0.76$). Raising $[Ca^{2+}]_o$ from 2 to 20 mM increased $^{55}Fe$ efflux for wildtype Fpn ($P < 0.001$) but not for either mutant ($P \geq 0.73$). All bar graphs indicate mean, SD, and individual scores; data in **b** are mean, SD

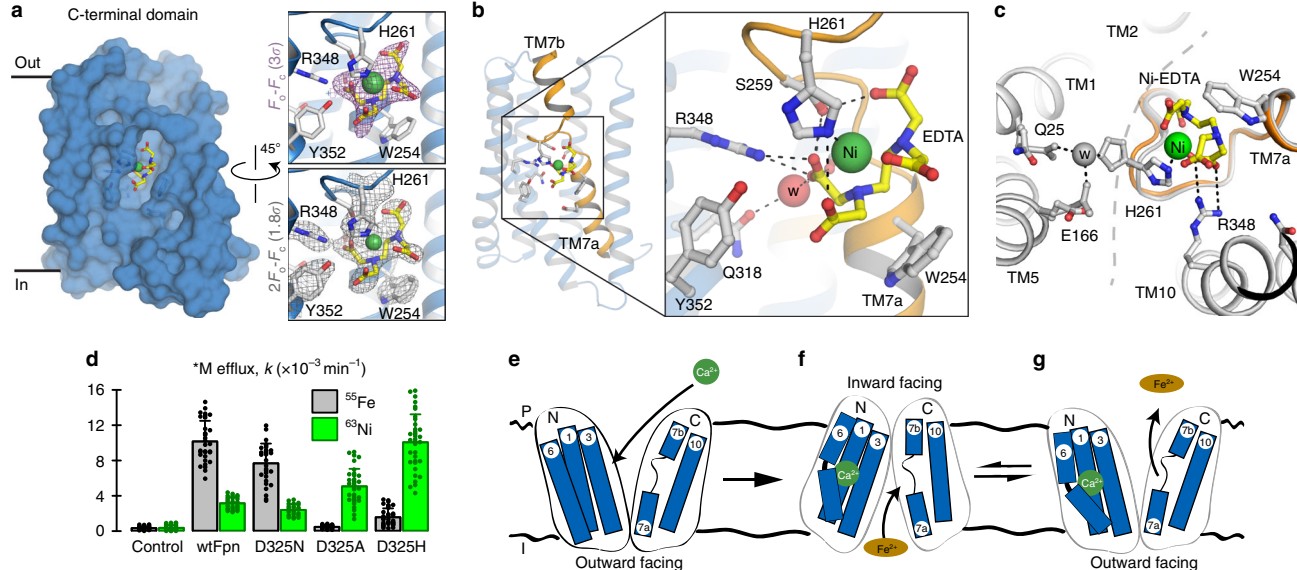

**Fig. 7** Putative substrate-binding site located in the C-terminal domain. **a** Surface representation of the C-terminal domain of BbFpn, viewed from the central cavity. The pocket formed by the non-continuous TM7 is in the center of the domain, with the Ni-EDTA molecule shown in ball-and-stick. Insets show $Fo$–$Fc$ omit (top) and $2Fo$–$Fc$ map (bottom) of the Ni-EDTA site. **b** Close-up view of the Ni-EDTA-binding site and coordination. Main hydrogen bonds are indicated by dotted lines. R348 forms a salt-bridge to one of the carboxylate groups of the EDTA moiety, shown in yellow, whereas H261 is a direct ligand to the metal (green sphere). **c** Superposition of the Ni-EDTA bound structure and the previously determined open inward conformation (gray; PDB entry 5AYO) illustrating conformational change of H261. **d** Effects of mutating residues within the putative metal-binding site of human Fpn. $^{55}Fe$ efflux (black) and $^{63}Ni$ efflux (green) from control oocytes and oocytes expressing wtFpn or one of several amino-acid substitutions at residue D325 ($n = 24, 29, 32, 26, 30, 22, 28, 35, 33, 35$, left to right). The bar graph indicates mean, SD, and individual scores. Two-way ANOVA revealed an interaction ($P < 0.001$). Within $^{55}Fe$ efflux, each mutant differed from wtFpn ($P < 0.001$); and D325N ($P < 0.001$) but neither D325A ($P = 0.78$) nor D325H ($P = 0.025$) differed from control. Within $^{63}Ni$, all groups differed from control ($P < 0.001$); D325A and D325H differed from wtFpn ($P < 0.001$) but D325N did not differ from wtFpn ($P = 0.14$). **e**–**g** Working model to illustrate $Ca^{2+}$-activated Fpn-mediated iron efflux

transporters, including proteins with the LeuT-type fold (e.g. LeuT and vSGLT)[20,21], nucleoside transporters[22], and other MFS-type transporters (e.g. Glut5 and MelB)[23,24]. In BbFpn, this pocket undergoes partial closure through a lateral shift of TM7b when comparing the outward and inward facing conformations, a shift reminiscent of that observed during substrate occlusion in sugar transporters[23]. However, this pocket was empty in the

structures previously obtained for BbFpn. Unexpectedly, in the pocket of the new structure, we observed clear non-protein electron density, which we have unambiguously assigned to a Ni-EDTA complex carried through from the protein purification procedure (Fig. 7a, b; see Methods).

In the structure, the EDTA moiety is seated in a largely hydrophobic area formed by residues from TM7, TM10, TM11,

and is hydrogen-bonded to the hydroxyl group of S256 and S259, carbonyl group of H261, and backbone amino groups of G262 and V263. In addition, R348, which is conserved in all Fpn proteins (Supplementary Fig. 2), forms a bidentate ligand to a carboxylate group of the EDTA molecule (Fig. 7b). Completing the coordination sphere, the $Ni^{2+}$ of the Ni-EDTA complex is directly coordinated by Nε of H261 (Ni–Nε 2.0 Å). The previous inward facing structure was determined at non-physiological pH (pH 5)[8], likely rendering H261 to the cationic imidazolium state, which is unable to coordinate metals due to repulsive charge (but able to act as hydrogen bond donor)[25]. In the $Ca^{2+}$-bound structure H261 changes conformation by 180° and forms a direct ligand to the metal in the Ni-EDTA complex (Fig. 7c).

Previous work has shown that both H261 and R348 are critical for metal binding in BbFpn[26]. Consistent with these observations, ITC measurements of wild type BbFpn gave an apparent dissociation constant of $K_d^{NiEDTA} = 0.84 \pm 0.07\,\mu M$ for Ni-EDTA, whereas the mutants H261A and R348A exhibited no observable affinity (Supplementary Fig. 9a–f). Although EDTA is not a physiologically relevant molecule, it possibly points towards a more complex transport model for the Fpn proteins in which they bind and transport metal in complex with an anion or a polycarboxylate metabolite, such as is the case for the CitMHS metal–citrate transporters[27,28].

To investigate a potential role of aspartate-325—the residue in human Fpn orthologous to the $Ni^{2+}$-coordinating H261 of BbFpn—in metal–substrate binding, we mutated it to asparagine (D325N), a substitution that modestly reduced $^{55}Fe$ efflux (Fig. 7d). Substituting D325 for alanine or histidine abolished $^{55}Fe$ efflux. In contrast, expression of D325A or, to a greater degree, D325H-stimulated $^{63}Ni$ efflux activity to levels exceeding that of wildtype Fpn (Fig. 7d and Supplementary Fig. 10). That certain amino-acid substitutions at D325 could convert the iron-preferring human Fpn into a nickel-preferring transporter provides compelling evidence for the involvement of D325 in metal binding and selectivity.

## Discussion

We have demonstrated that $Ca^{2+}$ is a required cofactor in Fpn-mediated metal efflux, stimulating iron efflux half-maximally at $[Ca^{2+}]_O = 0.8–0.9\,mM$. This observation raises the possibility that $Ca^{2+}$ binding in Fpn serves a regulatory function, and that the activity of Fpn could be limited in conditions of hypocalcemia —specifically, low plasma ionized calcium (i.e., $[iCa^{2+}] < 1.0\,mM$) characteristic of conditions such as primary hypoparathyroidism, chronic kidney disease, and hypomagnesemia (e.g. in the prolonged use of proton-pump inhibitors)[29,30].

Our results provide a framework for a transport model for the Fpn proteins (Fig. 7e–g). In the outward facing structure, the $Ca^{2+}$ site is solvent accessible. From this state, we propose that the binding of $Ca^{2+}$ from extracellular fluid activates Fpn by triggering a conformational change that enables the transition from the open outward to open inward states. In the open inward state, the protein can bind the metal substrate, possibly in complex with a carboxylate or anion, similar to that in our Ni-EDTA bound structure. The binding of the substrate triggers a return to the open outward conformation. In this state, TM7b, which contains the metal-binding residue, moves away from the substrate-binding pocket as seen in the open outward structure. We propose that the movement of TM7b promotes the release of the substrate by disrupting the metal–substrate coordination sphere (Supplementary Fig. 11a–c). An analogous structural change promoting substrate release can be observed in the mammalian plasma iron carrier protein transferrin (Supplementary Fig. 11d, e)[31]. Our findings thus advance our understanding of Fpn-mediated iron efflux and may aid in the development of strategies to manipulate Fpn therapeutically.

## Methods

**Reagents.** Reagents were obtained from Research Products International Corp. (Prospect, IL) or Sigma-Aldrich Corp. (St. Louis, MO) unless otherwise indicated.

**Expression of human Fpn in *Xenopus* oocytes.** We performed laparotomy and ovariectomy on adult female *Xenopus laevis* frogs (Nasco, Fort Atkinson, WI) under 2-aminoethylbenzoate methanesulfonate anesthesia (0.2%, by immersion, to effect) following a protocol approved by the University of Cincinnati Institutional Animal Care and Use Committee. Ovarian tissue was isolated and treated for 3 h with 2 mg l$^{-1}$ collagenase A (Roche Diagnostics Corp., Indianapolis, IN) in calcium-free-modified Barths' medium (MBM) of composition 88 mM NaCl, 1 mM KCl, 2.4 mM NaHCO$_3$, 1.57 mM MgSO$_4$, 0.66 mM NaNO$_3$, 10 mM 4-(2-hydroxyethyl)-1-piperazineethanesulfonic acid (HEPES), buffered to pH 7.5 by using 2-amino-2-hydroxymethyl-propane-1,3-diol (Tris base). We isolated defolliculate stage V–VI oocytes stored at 17 °C in MBM of composition 88 mM NaCl, 1 mM KCl, 2.4 mM NaHCO$_3$, 0.82 MgSO$_4$, 0.75 mM CaCl$_2$, 0.66 mM NaNO$_3$, 10 mM HEPES, buffered to pH 7.5 by using Tris base, with 50 mg l$^{-1}$ ciprofloxacin. We expressed in *Xenopus* oocytes the 1A transcript form of human Fpn fused at its C-terminus with the enhanced green fluorescent protein (GFP) as described[10]. Oocytes were injected with ~50 ng of Fpn RNA and incubated for 4–6 days before being used in functional assays. Control oocytes were noninjected.

**Radiotracer efflux assays in oocytes.** We measured metal efflux from oocytes expressing Fpn by using a radiotracer assay. Radiochemicals $^{55}Fe$, $^{45}Ca$, $^{57}Co$, $^{65}Zn$ were obtained from Perkin-Elmer Life Science Products (Boston, MA) and $^{63}Ni$ from the National Isotope Development Center (Oak Ridge, TN). We used $^{55}Fe$ (added as FeCl$_3$) at final specific activity 0.4–2.0 GBq mg$^{-1}$, $^{45}Ca$ (added as CaCl$_2$) at final specific activity 0.5–0.9 GBq mg$^{-1}$, $^{57}Co$ (added as CoCl$_2$) at final specific activity 1.6 GBq mg$^{-1}$, $^{65}Zn$ (added as ZnCl$_2$) at final specific activity 0.2 GBq mg$^{-1}$, and $^{63}Ni$ (added as NiCl$_2$) at final specific activity 0.6 GBq mg$^{-1}$. Oocytes were injected with 50 nl of 5 μM metal radionuclide (*M) in vehicle of composition 250 mM KCl, 5 mM nitrilotriacetic acid, trisodium salt, buffered to pH 7.0 with ~4 mM 2-(N-morpholino)ethanesulfonic acid (MES) (GFS Chemicals, Columbus, OH). After injection, oocytes were briefly stored in ice-cold MBM (for 1–5 min) prior to the start of the efflux assay.

To initiate the efflux assay, we placed oocytes in standard efflux medium of composition 100 mM NaCl, 1 mM KCl, 2 mM CaCl$_2$, 1 mM MgCl$_2$, 0.5 mM bathophenanthroline disulfonic acid (BPS), 1 mM nitrilotriacetic acid (NTA) trisodium salt, 50 μg/ml apotransferrin (ApoTf) (R&D Systems Inc., Minneapolis, MN), buffered with 0–5 mM MES, and 0–5 mM N′,N′-diethylpiperazine to obtain pH 7.5, with the following exceptions. To prepare $Ca^{2+}$-free medium, we replaced 2 mM CaCl$_2$ with 4 mM NaCl and added 1 mM ethylene glycol-bis(2-aminoethyl ether)-N, N,N′,N′-tetraacetic acid (EGTA). To prepare $Mg^{2+}$-free medium, we replaced 2 mM MgCl$_2$ with 4 mM NaCl and added 1 mM ethylenediaminetetraacetic acid (EDTA). To measure $^{55}Fe$ efflux as a function of extracellular $[Ca^{2+}]$ over the range 0.1–20 mM $Ca^{2+}$ (Figs. 1c, 6b), we adjusted [NaCl] and [CaCl$_2$] on a 2:1 molar basis without the addition of EGTA. Oocytes were then transferred individually in 200 μl of medium to flat-bottomed wells of a 96-well plate and incubated 30 min—within the linear phase of metal efflux[10]—at 23–27 °C with gentle rocking. To terminate efflux, we removed 150 μl of medium (0.75 × [$N_0$–$N_t$]) from each well for liquid-scintillation counting in Ultima Gold liquid-scintillation cocktail (Perkin-Elmer, Waltham, MA). The corresponding oocytes were rinsed twice in ice-cold medium (without BPS), solubilized with 5% (w/v) sodium dodecyl sulfate, and counted ($N_t$). We fit our data for oocyte radiotracer content ($N$) over time (Eq. (1)) and obtained the first-order rate constant ($k$) for metal radionuclide (*M) efflux for individual oocytes. $N_0$ and $N_t$ are counts per min (cpm) in the oocyte at time = 0 and time = t, respectively.

$$N_t = N_0 \cdot \exp(-k \cdot t) \qquad (1)$$

We took the first-order rate constants ($k$) of $^{55}Fe$ efflux as an index of transport velocity ($v$) and fit the data as a function of extracellular calcium concentration by a two-parameter hyperbolic function (Michaelis–Menten saturation) (Eq. (2)). $V_{max}$ is the maximal velocity, $C$ is the extracellular calcium concentration, and $K_{0.5}^{Ca}$ is the extracellular calcium concentration at which velocity was half-maximal.

$$v = (V_{max} \cdot C)/(K_{0.5}^{Ca} + C) \qquad (2)$$

**Radiotracer uptake assay.** We measured calcium uptake by using a radiotracer assay. We injected control oocytes and oocytes expressing Fpn with 50 nl of 50 μM FeCl$_3$ in a vehicle of composition 250 mM KCl, 5 mM NTA trisodium salt, buffered to pH = 7.0 by using MES, or vehicle only. After injection, oocytes were briefly stored in ice-cold MBM (for 1–5 min) prior to the start of the uptake assay. Oocytes were rinsed briefly with standard efflux medium and then placed in

medium of composition 100 mM NaCl or choline choride, 1 mM MgCl$_2$, 0.5 mM BPS, 1 mM NTA trisodium salt, 50 µg/ml ApoTf, buffered by using 0–5 mM MES and 0–5 mM DEPP to obtain pH 7.5 and containing 100 µM $^{45}$Ca for 30 min at 23 °C. We terminated radiotracer uptake by rapidly washing the oocytes three times in ice-cold medium (pH 7.5). Oocytes were prepared for liquid-scintillation counting as before.

**Functional expression of Fpn in HEK293T cells.** We measured $^{55}$Fe efflux from HEK293 cells (E. Nemeth, UCLA) that were stably transfected with a tet-inducible human Fpn construct in pcDNA5/FRT/TO vector (Invitrogen) as described[32]. Fpn expression was induced by adding 0.5 µg/ml doxycycline; data represent % efflux in induced minus uninduced cells. Fpn expression in mammalian cells was confirmed by visualizing GFP fluorescence, which was unaffected by removal of extracellular calcium.

**Site-directed mutagenesis.** Single mutations (D39E, D325A, D325N, D325H) were introduced into human Fpn in pcDNA5/FRT/TO by using QuikChange Lightning Site-Directed Mutagenesis Kit (Agilent Technologies) along with the primers listed in Supplementary Table 1, and the mutated hFpn–EGFP cDNA subcloned into the oocyte expression vector pOX(+). All other single and double mutations were generated by synthesis of a 901 bp mutant DNA containing each target mutation flanked by EcoRI and BlpI sites. The synthesized mutant DNA fragments were digested by EcoRI and BlpI and then subcloned into pOX(+) between EcoRI and BlpI to replace the wildtype fragment with mutant fragment (a service provided by Mutagenex, Suwanee, GA). All mutations were confirmed by sequencing.

**Imaging of oocytes expressing EGFP-tagged human Fpn mutants.** Oocytes in MBM were mounted on the Zeiss LSM 710 META confocal laser-scanning microscope (LSM). GFP fluorescence was visualized by using the C-45 Apochromat ×10/0.45 W objective, exciting at 488 nm, and detecting fluorescence in the range 505–530 nm with optical slice of ≈8 µm approximately bisecting the oocyte.

**Oocyte and HEK cell data analysis.** We performed two-tailed statistical analyses of functional data from oocyte and HEK cell preparations by using SigmaPlot version 13 (Systat Software) with critical significance level $\alpha = 0.01$. We have presented our data as mean and standard deviation (SD) for $n$ independent observations, except in Fig. 4 in which individual scores are presented, and used parametric tests in their analysis since the data met assumptions of normality and homoscedasticity. Data in Figs. 1c, 6b were fit by Eq. (2) by using least-squares regression analysis, the results of which are expressed as the estimates of fit parameters ± standard error of the mean (SEM); $r^2$ is the regression coefficient and $P$ describes the significance of the fit. We tested between-group comparisons using two-way analysis of variance (ANOVA), with repeated measures (RM) on both factors in the case of Fig. 4, followed by pairwise multiple comparisons using the Holm–Šídák test where appropriate. We estimated minimum sample sizes required to achieve desired power 0.9 (i.e. $\beta = 0.1$) to detect a minimum pre-specified effect (of relative size $\delta$) in each of the following three generalized experimental designs: (1) two-way ANOVA (ion dependence, Ca$^{2+}$ transport, mutant activity in oocytes): $\delta = 0.5$ (e.g. 50% inhibition) and coefficient of variation, $\gamma = 0.25$, minimum sample size required $n = 7$. (2) Regression analysis: $r^2 = 0.8$, $\gamma = 0.25$, $n = 9$. (3) Two-way RM ANOVA (Ca$^{2+}$ dependence of Fe transport in HEK cells): $\delta = 0.5$, $\gamma = 0.2$, $n = 4$. Findings from oocyte expression data presented were replicated in at least two additional preparations.

**BbFpn gene cloning and protein expression.** The gene for BbFpn with flanking XhoI and BamHI restriction sites was purchased from GenScript (USA) and cloned into pWaldoE[33], generating a C-terminal fusion with GFP-His8 and intermittent tobacco etch virus (TEV) protease cleavage site. Protein was expressed in E. coli C41 cells[34] (G. von Heijne, Stockholm University) grown in terrific broth (TB) at 37 °C until OD$_{600}$ reached 0.8, after which the temperature was reduced to 20 °C. Expression was induced by adding 0.2 mM IPTG, and cells were harvested by centrifugation after ~8 h of expression and resuspended in buffer containing 20 mM Tris pH 8 and 100 mM NaCl before freezing at −20 °C until further use.

**Fpn protein expression and purification.** To obtain a purified mammalian Fpn protein conducive for biophysical studies, we generated a protein construct with an increased thermal stability (Deshpande et al., 2018). In brief, the construct (Fpn-C2) was generated from full-length mouse Fpn (UniProt: Q9JHI9 [https://www.uniprot.org/uniprot/Q9JHI9]) by deleting two-loop regions (residues 251–290 and 401–449), followed by an overlapping PCR incorporating the E219A mutation to generate Fpn-C2-E219A. The genes were subsequently cloned with a C-terminal GFP fusion into a baculovirus transfer vector (pFastBac1; Thermo Fisher Scientific) and the proteins were expressed using the Bac-to-Bac Baculovirus and Sf9 cells (Thermo Fisher Scientific) expression method according to the manufacturer's protocol (Thermo Fisher Scientific). For large scale expression of the protein, 6L Sf9 cells (2 × 10$^6$ cells/ml) were infected with virus at an optimal multiplicity of infection (MOI) of 2.5 in Insect-XPRESS™ Protein-free Insect Cell Medium with

L-glutamine (Lonza) and incubated at 27 °C with shaking at 130 rpm. The cells were spun down 48 h post-infection at 1500 × g for 10 min and the pellet stored at −20 °C until further use.

**Protein purification for ITC studies.** For BbFpn protein purification, cells were thawed at room temperature with 0.2 mM phenyl methyl sulfonyl fluoride (PMSF) and lysed by three passes through a cooled EmulsiFlex-C3 homogenizer (Avestin). All subsequent steps were at 4 °C. Membranes were harvested by centrifugation at 100,000 × g for 75 min, and pellets resuspended in buffer containing 20 mM Tris pH 8 and 300 mM NaCl (Buffer A). Membrane proteins were solubilized by the addition of 1% (final concentration) n-dodecyl-β-D-maltoside (DDM) and incubated under stirring for one hour. Insoluble cellular debris was subsequently cleared by centrifugation at 80,000 × g for 20 min. The supernatant from this was thereafter incubated with Ni-NTA (Qiagen), pre-equilibrated with Buffer A, for 2 h under slow stirring. The protein-bound resin was loaded onto a Poly-prep gravity flow column (BioRad) and the resin was subsequently washed with 10 column volumes of 20 mM imidazole pH 8, 300 mM NaCl, and 0.03% DDM (Buffer B). Protein was then eluted with buffer containing 200 mM imidazole, 300 mM NaCl, and 0.004% lauryl maltose neopentyl glycol (LMNG). The GFP moiety was cleaved from the eluted protein using TEV protease (containing 0.5 mM EDTA) under dialysis with 20 mM Tris, pH 8, 300 mM NaCl (overnight). The GFP moiety and TEV protease were cleared from BbFpn by rebinding dialyzed protein to Ni-NTA. The untagged protein was eluted and concentrated to 1.8 ml before loading onto a Superdex 200 16/600 size exclusion column (GE Healthcare Life Sciences), pre-equilibrated with 20 mM Tris pH 8, 50 mM NaCl, and 0.004% LMNG (gel filtration buffer A). The eluted protein was concentrated to ~10 mg/ml as determined by the BCA assay (Thermo Fisher Scientific) method and stored at −80 °C until further use.

For Fpn-C2 and Fpn-C2-E219A purification (Deshpande et al., 2018), Sf9 cells were thawed at room temperature with the addition of 2 µl Benzonase (250 units/µL, Sigma-Aldrich), 1 mM MgSO$_4$, 0.2 mM PMSF, and lysed by a single pass through cooled EmulsiFlex-C3 homogenizer (Avestin). All subsequent steps were performed at 4 °C. Membranes were harvested by centrifugation at 106,000 × g for 1 h, and pellets resuspended in Buffer A. The protein was subsequently purified as BbFpn, but with the addition of a second wash on the Ni-NTA column with three column volumes of 25 mM imidazole pH 8, 300 mM NaCl, and 0.03% DDM, and with 0.03% DDM instead of LMNG in the elution and SEC purification. The SEC eluted protein was concentrated to ~10 mg/ml as determined by the BCA assay (Thermo Fisher Scientific) method and stored at −80 °C until further use.

**Protein expression and purification for crystallization.** For the crystallographic analysis, the protein sample of BbFpn was prepared in a similar manner as described above, with slight modifications. The same pWaldoE vector was used for the expression, and 10 residues at the C terminus of BbFpn were deleted. Protein was expressed in E. coli C41 cells[34] (G. von Heijne, Stockholm University) grown in Luria broth (LB) medium. Transformed cells were grown at 37 °C to an OD$_{600}$ of 0.9, and protein expression was induced with 0.2 mM IPTG at 20 °C for 18 h. The cells were harvested and resuspended in buffer containing 20 mM Tris pH 8.0, 300 mM NaCl, and 0.1 mM PMSF, and subsequently disrupted by three passages at 15,000 p.s.i. through a Microfluidizer processor (Microfluidics). The debris was removed by centrifugation (12,500 × g, 30 min, 4 °C), and the supernatant was further ultracentrifuged (125,000 × g, 1 h, 4 °C) to collect the membrane fraction.

The membrane pellet was solubilized in buffer containing 20 mM Tris pH 8.0, 300 mM NaCl, 10 mM imidazole pH 8.0, 0.1 mM PMSF, and 1% DDM, and stirred for 2 h at 4 °C. Insoluble cellular debris was removed by ultracentrifugation (125,000 × g, 1 h, 4 °C), and the supernatant was incubated with Ni-NTA (Qiagen), pre-equilibrated with buffer containing 20 mM Tris pH 8.0, 300 mM NaCl, 20 mM imidazole pH 8.0, and 0.03% DDM (equilibration buffer), for 1 h under slow stirring. The resin was washed using 10 column volumes of the equilibration buffer, and then further washed with five column volumes of similar buffer containing 0.01% LMNG instead of DDM. Protein was then eluted with buffer containing 20 mM Tris pH 8.0, 300 mM NaCl, 200 mM imidazole pH 8.0, and 0.01% LMNG. The GFP was cleaved from the eluted protein using His-tagged TEV protease (containing 0.5 mM EDTA) under dialysis with buffer containing 20 mM Tris pH 8.0 and 300 mM NaCl (overnight). The GFP moiety and TEV protease were cleared from BbFpn by rebinding the dialyzed protein to Ni-NTA (i.e. re-binding of His-tagged GFP and TEV). The untagged BbFpn protein (including EDTA) was eluted and concentrated to 0.5 ml before loading onto a Superdex 200 16/600 size exclusion column (GE Healthcare Life Sciences), pre-equilibrated with 20 mM Tris pH 8.0, 300 mM NaCl, and 0.004% LMNG. The purified BbFpn was concentrated to ~15 mg/ml and subsequently used for crystallization.

**Protein crystallization.** The purified BbFpn was reconstituted into the lipidic cubic phase (LCP) by mixing with monoolein at a 2:3 protein to lipid ratio (w/w), using the twin-syringe mixing method[35]. After the reconstitution, 80 nl LCP drops were dispensed onto either LCP screening plates (SWISSCI), and overlaid with 800 nl of precipitant solution using the Mosquito LCP crystallization robot (TTP LabTech). Initial crystallization trials were performed using the MemMeso crystallization screen (Molecular Dimensions), and all the crystallization screenings

were performed at 20 °C. The final crystal was obtained in the precipitant solution containing 100 mM Tris pH 8.0, 100 mM Na-acetate, 70 mM $CaCl_2$, and 20% PEG500DME. The crystal was directly harvested from the crystallization plate and flash frozen in liquid nitrogen.

**Structural data collection and structure determination.** The diffraction data set was collected at SPring-8 BL32XU, using a micro-focused X-ray beam[36], and processed using the program XDS[37]. The structure was determined by molecular replacement with the program PHASER[38], using the N and C lobes of the previously determined BbFpn outward-facing structure as the search models. The model was refined using the program PHENIX[39], and manually rebuilt using COOT[40]. The data collection and refinement statistics are summarized in Table 1. All molecular graphics were prepared using PyMol (www.pymol.org).

**Isothermal titration calorimetry.** Protein samples were purified as above and diluted to 0.03 mM (1.5 mg/ml) for ITC experiments. The buffer used for ITC was 20 mM Tris pH 8, 50 mM NaCl, and 0.004% LMNG. The buffer was degassed in vacuum (2 h) and pretreated with Chelex-100 (Bio-Rad) to remove residual calcium and/or metal ions bound with moderate affinity. This was done by first washing the Chelex-100 beads with water and then running the dialysis buffer through the beads in a gravity flow column. The samples were subsequently dialyzed overnight against their respective gel filtration buffers. ITC measurements were obtained by using a MicroCal iTC 200 calorimeter (GE Healthcare) and MicroCal PEAQ-ITC (Malvern Instruments) at 20 °C. For calcium and strontium binding measurements, a 2 mM solution was prepared in the protein buffer using a standard 1 M stock solution. Ni-EDTA solution (0.6 mM) was prepared using equimolar concentrations of $Ni^{2+}$ and EDTA, pH 8 from a 1 M stock solution, and loaded on to the syringe. For $MgCl_2$ and KCl measurements, a 2–5 mM solution was prepared in the respective buffer.

In each experiment, the protein solution was loaded into the sample cell, and the ligand solution was loaded in the syringe. The reference cell was filled with the respective degassed buffer. The ligand was injected 25 times (first injection of 0.2 μl followed by 24 injections at 1 μl), with a stirring speed of 750 rpm and 200 s intervals between injections. To obtain the effective heat of binding, the heat of dilution, measured by injecting the ligand solution alone into buffer, was subtracted prior to data analysis. The data were fitted using the 'one binding site' model using the MicroCal PEAQ-ITC Analysis Software (Malvern Instruments), and the apparent molar reaction entropy ($\Delta H°$), entropy ($\Delta S°$), dissociation constant ($K_d$), and stoichiometry of binding ($N$) were determined. Our experiments were conducted a minimum of three times to ensure reproducibility. In the figures, we show the results of one experiment without statistical analysis.

**Data availability.** Data supporting the findings of this manuscript are available from the corresponding authors upon reasonable request. X-ray structural data and structure factors have been deposited in the Protein Data Bank under accession number 6BTX.

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

## Acknowledgements

We thank John P. Bonamer, Justin L. Dunham, Austin Jiang and Shahana Prakash (University of Cincinnati) for their assistance in the laboratory, and Hideaki Kato and Reiya Taniguchi (University of Tokyo) for assistance with X-ray data collection and model building. This study was supported primarily by National Institute of Diabetes and Digestive and Kidney Diseases (NIDDK) grant R01 DK107309 (to M.J., B.M., and E.N.). Additional support was provided by the National Health and Medical Research Council (NHMRC) (grant no. APP1083536) and Australian Cancer Research Fund (to M.J.); NIDDK grants R01 DK080047 (to B.M.), R01 DK082717 (to E.N.), and P30 DK078392 (Digestive Health Center, Cincinnati Children's Hospital and University of Cincinnati); and the University of Cincinnati (to B.M.). The content of this study is solely the responsibility of the authors and does not necessarily represent the official views of the NIDDK, the National Institutes of Health, and NHMRC.

## Author contributions

B.M. and M.J. conceptualized the study. C.N.D., T.A.R., A.S., V.X., K.R.V., E.V.V., B.Q., E.N. T.G., B.M., and M.J. designed and performed experiments, and analyzed the data. C. N.D., T.A.R., B.M., and M.J. performed statistical analyses. C.N.D., T.A.R., B.M., and M.J. wrote the manuscript. All authors edited the manuscript and approved the final version.

## Additional information

**Competing interests:** T.G. and E.N. are scientific advisers to and shareholders in Intrinsic LifeSciences and Silarus Therapeutics, and consultants for La Jolla Pharmaceutical Company and Keryx Pharmaceuticals. B.M. is a grant recipient of Vifor Pharma. The remaining authors declare no competing interests.

