## [Peer Review File · Nature Communications]

Reviewers' comments:

Reviewer #1 (Remarks to the Author):

The authors report that calcium is required for human ferroportin transport activity. Based on crystal structure data of a prokaryotic ortholog of ferroportin they conclude that calcium is a cofactor that facilitate a conformational change that may be required for efficient iron export. This paper yields novel insight into aminoacids in human ferroportin required for iron transport and a role of calcium as a cofactor

Reviewer #2 (Remarks to the Author):

In this manuscript, the authors report a comprehensive study of the calcium binding properties of ferroportin and the modulatory effect of cation binding on the transport activity of the protein. To this purpose, they continuously switch from data on the purified and crystallized bacterial ferroportin from *Bdellovibrio bacteriovorus* to data on the human protein expressed in two heterologous expression systems (oocytes, HEK cells).

The paper is of value and data may deserve to be published. However, a number of points need to be addressed/clarified in order for the manuscript to be reevaluated for acceptance.

Major points

1. The authors demonstrate that calcium is not transported by human Fpn, however calcium binding is directly demonstrated only on the bacterial protein.
2. (following point 1) The authors show that mutation of the putative Ca²⁺ coordinating residues impairs calcium binding in BbFpn, and that mutation of the orthologous residues in human Fpn impairs iron transport. While it is reasonable to assume that a sort of transitive property can be applied, they should complete the set of experiments by running iron transport assays in bacteria expressing WT and mutated BbFpn, as well as calcium binding measurements in purified human Fpn (WT and mutants).
3. The authors claim that the previously assigned substrate site in BbFpn is not of physiological interest as it would be filled only at very high concentrations of cations. However, it is not clear why the "right" metal binding site did not show up in the crystal soaked with iron, as one would expect the physiological site to bind iron with higher affinity with respect to the non-physiological site. A possible reason is that the Fe-soaked crystal had BbFpn in the open outward conformation, likely with a loosened iron binding site, but the authors should better discuss this point.
4. When comparing Figg. 1B and 4B, there is a significant discrepancy between V_{max} of iron transport in oocytes transfected with WT Fpn. Please explain.
5. As the authors state, EDTA is not a physiological ligand for iron, yet they base their structural conclusions on the precise fit of the Ni-EDTA complex into the substrate binding site of Fpn. Docking of physiological complexes like Fe-citrate could be attempted to show that they can equally fit into the substrate pocket. The authors should also tell whether any attempt of crystallization of BbFpn in the presence of other iron or nichel complexes has been made.

Minor points:

1. The authors say that calcium dependence for zinc transport by human Fpn was not tested, but do not explain why.
2. Supplementary Fig. 1 does not report the right panels (it instead reproduces Fig. 1 of the main text).
3. To be consistent with the text, Fig. 2C should also show TM6 in the inward open conformation in the absence of calcium.
4. The previously assigned substrate site also involves Ser199, which was not mentioned in the text.

Reviewer #3 (Remarks to the Author):

Review of NCOMMS-18-00294-T

Comments to authors:

The manuscript submitted by Despanthe and coworkers describes a series of studies pertaining to the activity of the ferroportin iron-transporter. This is currently the only known iron-efflux protein. The authors demonstrate quite convincingly that addition of calcium (Ca^{2+}) promotes iron transport. They study the effects of various other metal ions, and conclude that the related strontium ion is the only other divalent cation that can mimick the effects of calcium. Apart from carrying out transport studies the authors also performed direct binding ITC studies, which provided information that is consistent with the previous transport data. They then go on and determine the crystal structure of a bacterial ferroportin homolog and locate the calcium binding site in the protein. Subsequently they generate a series of mutants in the calcium ligands and the results are again in line with the expectations. A final piece of the work centers on the unexpected binding of nickel-EDTA to the protein. The authors argue that this provides some insight into the location of the iron-binding/transport site. The paper is generally well written, but it is very condensed and sometimes a little tough to follow. I guess, to some extent, this reflects the typical format of the Nature Communications journal. The statistical data seem to be quite well addressed. The strength of the paper lies in the calcium-binding aspects of the protein; I find the nickel-EDTA observations a bit more mysterious. I have the following questions and comments:

- 1) There is an error in the numbering on the title page. The 6 superscript is being used as the locator for the address of one of the authors (Ganz), but also to indicate that the first two authors contributed equally to the work. This needs to be fixed, there should be separate symbols. Moreover there is a typo here: "These authors...."
- 2) Comment: The original pdf file in the system had 32 pages but did not have the supplementary figures. These were found in a separate PDF document on the journal website. In the first document the methods were placed after the literature references, which seems odd to me, but may reflect journal policy. A more serious comment is that Figure 1 in the manuscript and the supplementary figure 1 seem completely identical to me in the files provided. What is the difference?
- 3) It would have been much better to get an acceptance report confirming the quality of the new structures from the Protein Databank before the paper was send out for review. Many journals now request this upfront, and I recommend that this paper should not be accepted until this is done (and a PDB code is provided).
- 4) The authors seem to suggest in the text that strontium has essentially the same ionic radius as calcium. This is incorrect, it is in fact somewhat larger than calcium and this should be mentioned as a factor to help explain the weaker binding of strontium compared to calcium. It is of interest that the magnesium and barium divalent cations don't seem to bind. Barium is even bigger than strontium and apparently does no longer fit in the site. Magnesium is smaller than calcium, so it could potentially fit, but it is possible that it does not bind, because it hangs on to its water molecules quite tightly(compared to calcium). The authors gloss over these aspects in the ms.
- 5) Although the new protein structure is apparently closed from the outside, the authors seem to suggest that the calcium enters from the outside environment, where the calcium concentration is high (around 1 mM); this is consistent with the binding constant determined. Looking at the structure, the calcium is quite deep into the protein and seems to be located closer to the inside environment. Could it possibly have entered from that side? I agree, it would seem unlikely given

the known binding constant and the low calcium concentrations inside the cell. I guess the point of how the bound calcium may get to its position, should be more clearly addressed in the paper, to make it clearer for the reader.

6) The ITC binding data seem of good quality and show endothermic binding of calcium. In the case of typical EF-hand calcium binding proteins the binding of calcium is usually exothermic. Clearly the calcium-binding site here must be quite different from the typical EF-hand sites. Having said that, the proposed coordination as a tetrahedral pyramid seems odd. Most high-resolution protein calcium-binding sites published to date seem to display 7-coordination in a pentagonal bipyramid arrangement, hence it seems like something is missing here. Are there maybe coordinating water molecules missing here that could further fill out the binding site? Perhaps the resolution in this part of the structure is not good enough to draw detailed conclusions about the calcium coordination?

7) If I understand this correctly, the new structure was determined at pH 8, while previous structures of the same protein were done at pH5. If this is correct, this should be mentioned directly in the text in the results section, to make it easier for the reader to follow the subsequent discussion on the roles of histidine, etc.

8) Page 4. Why was zinc transport not tested, given that the authors mention that it is a known ligand for transport? Likewise why was nickel transport tested here, even though it is apparently not a known ligand (at least it was not mentioned by the authors)?

9) I am confused about the appearance of the nickel-EDTA complex in the structure. His-tagged proteins are usually washed of the nickel-NTA columns with high concentrations of imidazole. I see in the methods that a TEV cleavage step was used and this does indeed require EDTA, but this apparently was followed by dialysis and then repurification over nickel-NTA. Hence it would seem that the EDTA was removed and from this procedure it seems more likely that nickel imidazole would be found bound to the protein. This raises a number of questions, is the EDTA density conclusive, or could it have been an imidazole complex instead? Does the protein crystallize in the same manner with its tag attached? If that were the case TEV and EDTA would not be needed during the protein preparation and there likely should be no bound nickel-EDTA complex found. Finally have such 'contaminating' nickel-EDTA complexes been observed before in other published protein structures; if so this should be commented on in the supplementary material (given that His-tags and such nickel columns are in extremely widespread use). If not commonly seen, is this an anomalous observation and why do the authors feel that it happens here?

We would like to thank the reviewers for their critical reading of the manuscript and helpful comments. We were pleased to see that the reviewers appreciated the significant insights this work provides into understanding the molecular mechanism of Fpn-mediated iron transport. We have revised the manuscript in response to the reviewers' comments, and have included several additional biophysical experiments, which further support our conclusions and clarify the points in the paper. We have below responded to the points raised by the reviewers.

As the reviewers may be aware, our Ms was referred to Nat Commun after review at another Nature family journal, so our manuscript was not organized in the format expected for Nat Commun. We apologize to the reviewers for the inconvenience. We have expanded the Ms where necessary to fully address the reviewers' comments.

Reviewer #1

The authors report that calcium is required for human ferroportin transport activity. Based on crystal structure data of a prokaryotic ortholog of ferroportin they conclude that calcium is a cofactor that facilitate a conformational change that may be required for efficient iron export. This paper yields novel insight into aminoacids in human ferroportin required for iron transport and a role of calcium as a cofactor

Reviewer #2 (Remarks to the Author):

In this manuscript, the authors report a comprehensive study of the calcium binding properties of ferroportin and the modulatory effect of cation binding on the transport activity of the protein. To this purpose, they continuously switch from data on the purified and crystallized bacterial ferroportin from *Bdellovibrio bacteriovorus* to data on the human protein expressed in two heterologous expression systems (oocytes, HEK cells). The paper is of value and data may deserve to be published. However, a number of points need to be addressed/clarified in order for the manuscript to be reevaluated for acceptance.

Major points

1. The authors demonstrate that calcium is not transported by human Fpn, however calcium binding is directly demonstrated only on the bacterial protein.
2. (following point 1) The authors show that mutation of the putative Ca²⁺ coordinating residues impairs calcium binding in BbFpn, and that mutation of the orthologous residues in human Fpn impairs iron transport. While it is reasonable to assume that a sort of transitive property can be applied, they should complete the set of experiments by running iron transport assays in bacteria expressing WT and mutated BbFpn, as well as calcium binding measurements in purified human Fpn (WT and mutants).

We believe it reasonable, as the reviewer suggests, to assume that the BbFpn and Fpn proteins to have analogous Ca²⁺ binding sites. We also agree that it would be highly beneficial to show direct binding to a mammalian Fpn protein in solution. As such, we have in the past few months been working on obtaining purified mammalian Fpn protein for biophysical characterization. Due to the extent of this additional material, we have prepared a separate stand-alone manuscript describing the cloning, expression, and purification of mouse Fpn (draft manuscript attached). Given that ITC studies are extremely protein consuming and not generally feasible for mammalian membrane proteins (due to cost of production, low expression levels, poor stability in solution), we also optimized the construct for stability in solution.

We have obtained sufficient quantities of a thermostabilized protein construct of mouse Fpn (Fpn-C2) and a Fpn-C2-E219A mutant for ITC studies. We have added to the manuscript the ITC analysis of

these proteins. The analysis illustrates similar endothermic affinity for Ca^{2+} by Fpn-C2 ($K_d^{\text{Ca}} \sim 3 \mu\text{M}$) as the BbFpn protein, whereas the Fpn-C2-E219A has no binding.

These results have been included in the revised manuscript:

*“By using ITC with purified thermostabilised mouse Fpn protein (Fpn-C2; Deshpande et al, 2018), we confirmed Ca^{2+} binding also in the mammalian ortholog ($K_d^{\text{Ca}} = 2.7 \pm 0.7 \mu\text{M}$; **Supplementary Fig. 6A**), whilst we observed no binding to purified Fpn-C2-E219A mutant protein (**Supplementary Fig. 6B**).”*

In ‘Methods’:

“Fpn protein expression and purification

To obtain a purified mammalian Fpn protein conducive for biophysical studies, we generated a protein construct with an increased thermal stability (Deshpande et al, 2018). In brief, the construct (Fpn-C2) was generated from full-length mouse Fpn (UniProt: Q9JHI9) by deleting two loop regions (residues 251-290 and 401-449), followed by an overlapping PCR incorporating the E219A mutation to generate Fpn-C2-E219A. The genes were subsequently cloned with a C-terminal GFP fusion into a baculovirus transfer vector (pFastBac1; Thermo Fisher Scientific) and the proteins were expressed using the Bac-to-Bac Baculovirus expression method according to the manufacturer’s protocol (Thermo Fisher Scientific). For large scale expression of the protein, 6L Sf9 cells (2×10^6 cells/ml) were infected with virus at an optimal multiplicity of infection (MOI) of 2.5 in Insect-XPRESS™ Protein-free Insect Cell Medium with L-glutamine (Lonza) and incubated at 27 °C with shaking at 130 rpm. The cells were spun down 48h post-infection at 2,500 rpm for 10 min and the pellet stored at -20 °C until further use.”

“Fpn-C2 and Fpn-C2-E219A were purified as described in Deshpande et al. (2018). In brief, Sf9 cells were thawed at room temperature with the addition of 2 μL Benzonase (250 units/ μL , Sigma-Aldrich), 1 mM MgSO_4 , 0.2 mM PMSF, and lysed by a single pass through cooled EmulsiFlex-C3 homogenizer (Avestin). All subsequent steps were performed at 4 °C. Membranes were harvested by centrifugation at $106,000 \times g$ for 1 hour, and pellets resuspended in Buffer A. The protein was subsequently purified as BbFpn, but with the addition of a second wash on the Ni-NTA column with 3 column volumes of 25 mM imidazole pH 8, 300 mM NaCl, and 0.03% DDM, and with 0.03% DDM instead of LMNG in the elution and SEC purification. The SEC eluted protein was concentrated to ~ 10 mg/mL as determined by the BCA assay (Thermo Fisher Scientific) method and stored at -80 °C until further use.”

Transport studies in bacterial cells have proven to be technically challenging, in part due to the presumed transport direction (efflux). However, we trust the ITC analysis of Ca^{2+} binding to mammalian Fpn sufficiently addresses the reviewers request.

3. The authors claim that the previously assigned substrate site in BbFpn is not of physiological interest as it would be filled only at very high concentrations of cations. However, it is not clear why the "right" metal binding site did not show up in the crystal soaked with iron, as one would expect the physiological site to bind iron with higher affinity with respect to the non-physiological site. A possible reason is that the Fe-soaked crystal had BbFpn in the open outward conformation, likely with a loosened iron binding site, but the authors should better discuss this point.

We believe there are two possible factors explaining this; (1) our structure with a bound Ni-EDTA complex hints at the possibility that the FPN proteins require a metabolite in complex with Fe to bind to the substrate-binding site. In the absence of the metabolite, Fe or other metals may, at high

concentration, bind only non-specifically to the protein (such as in soaking experiments). (2) As the reviewer suggests, the metal affinity in the outward facing conformation would be expected to have a much lower affinity. In our structural analysis, this may be rationalized by the conformational change of TM7b and the metal coordinating residue (H261) when comparing inward and outward facing structures; in the inward facing conformation, TM7b is positioned 'down', which enables H261 to coordinate the metal. In the outward facing structure, TM7b and H261 are positioned several Å further away from the metal site, effectively opening the 'substrate pocket' up, which we propose disrupts the metal binding coordination sphere. This may form part of a substrate release mechanism. To clarify this for the reader, we have added a **Supplementary Fig. 10**.

4. When comparing Fig. 1B and 4B, there is a significant discrepancy between V_{\max} of iron transport in oocytes transfected with WT Fpn. Please explain.

Data in what are now Fig. 1C and 5B are derived from two different oocyte preparations. Protein expression levels vary markedly between independent oocyte preparations (this is true of many proteins, not just Fpn) so we do not expect the transport V_{\max} to be the same. Moreover, a comparison of V_{\max} between preparations is not of interest, only the comparison of wildtype and Q99A within the same preparation (Fig. 5B).

5. As the authors state, EDTA is not a physiological ligand for iron, yet they base their structural conclusions on the precise fit of the Ni-EDTA complex into the substrate binding site of Fpn. Docking of physiological complexes like Fe-citrate could be attempted to show that they can equally fit into the substrate pocket. The authors should also tell whether any attempt of crystallization of BbFpn in the presence of other iron or nickel complexes has been made.

The pocket in which the Ni-EDTA molecule is situated is large enough to fit citrate or any of a number of possible chelators or metabolites. As such, we believe docking experiments will have little value and we will seek to identify any physiological co-substrate via direct methods instead (as proposed, using co-crystallization, ITC experiments, or transport studies). Nevertheless, whereas this could turn out to be a significant finding, we believe that it falls outside the main scope of this manuscript which describes activation by Ca, identification of the Ca-binding site, and subsequent reassignment of the metal-binding site. Ongoing and future work is aimed at investigating the hypothesis of a cotransport with a chelator or organic anion.

Minor points:

1. The authors say that calcium dependence for zinc transport by human Fpn was not tested, but do not explain why.

We now demonstrate that human Fpn requires extracellular calcium for the transport of iron, cobalt, nickel, and zinc (Fig. 1).

2. Supplementary Fig. 1 does not report the right panels (it instead reproduces Fig. 1 of the main text).

We apologize for this error, when compiling the supplementary file the wrong figure was inserted. Elements from Supplementary Fig. 1 now appear in Fig. 1 and Fig. 2.

3. To be consistent with the text, Fig. 2C should also show TM6 in the inward open conformation in the absence of calcium.

We have changed **Fig. 2C** (now Fig. 4C) according to the reviewers suggestion.

4. The previously assigned substrate site also involves Ser199, which was not mentioned in the text.

We have added S199 to the relevant section in the manuscript.

pg 7: “The latter site, comprised of T20, D24, N196, **S199**, and F200...”

Reviewer #3 (Remarks to the Author):

Review of NCOMMS-18-00294-T

Comments to authors:

The manuscript submitted by Despardhe and coworkers describes a series of studies pertaining to the activity of the ferroportin iron-transporter. This is currently the only known iron-efflux protein. The authors demonstrate quite convincingly that addition of calcium (Ca²⁺) promotes iron transport. They study the effects of various other metal ions, and conclude that the related strontium ion is the only other divalent cation that can mimic the effects of calcium. Apart from carrying out transport studies the authors also performed direct binding ITC studies, which provided information that is consistent with the previous transport data. They then go on and determine the crystal structure of a bacterial ferroportin homolog and locate the calcium binding site in the protein. Subsequently they generate a series of mutants in the calcium ligands and the results are again in line with the expectations. A final piece of the work centers on the unexpected binding of nickel-EDTA to the protein. The authors argue that this provides some insight into the location of the iron-binding/transport site. The paper is generally well written, but it is very condensed and sometimes a little tough to follow. I guess, to some extent, this reflects the typical format of the Nature Communications journal. The statistical data seem to be quite well addressed. The strength of the paper lies in the calcium-binding aspects of the protein; I find the nickel-EDTA observations a bit more mysterious. I have the following questions and comments:

1) There is an error in the numbering on the title page. The 6 superscript is being used as the locator for the address of one of the authors (Ganz), but also to indicate that the first two authors contributed equally to the work. This needs to be fixed, there should be separate symbols. Moreover there is a typo here: “These authors...”

This has been amended in the revised manuscript.

2) Comment: The original pdf file in the system had 32 pages but did not have the supplementary figures. These were found in a separate PDF document on the journal website. In the first document the methods were placed after the literature references, which seems odd to me, but may reflect journal policy. A more serious comment is that Figure 1 in the manuscript and the supplementary figure 1 seem completely identical to me in the files provided. What is the difference?

We apologize for this error, when compiling the supplementary file the wrong figure was inserted. Elements from Supplementary Fig. 1 now appear in Fig. 1 and Fig. 2.

3) It would have been much better to get an acceptance report confirming the quality of the new structures from the Protein Databank before the paper was send out for review. Many journals now request this upfront, and I recommend that this paper should not be accepted until this is done (and a PDB code is provided).

The validation report for the structure has been submitted to the Journal. The structure has also been approved by PDB and is currently on hold for publication.

4) The authors seem to suggest in the text that strontium has essentially the same ionic radius as calcium. This is incorrect, it is in fact somewhat larger than calcium and this should be mentioned as a factor to help explain the weaker binding of strontium compared to calcium. It is of interest that the magnesium and barium divalent cations don't seem to bind. Barium is even bigger than strontium and apparently does no longer fit in the site. Magnesium is smaller than calcium, so it could potentially fit,

but it is possible that it does not bind, because it hangs on to its water molecules quite tightly (compared to calcium). The authors gloss over these aspects in the ms.

We agree with the reviewer in that this can be explained in more detail. As such, we have added the following:

“To further characterize the cation specificity, we measured ^{55}Fe efflux in the presence of other alkaline-earth metals (in Ca^{2+} -free media) and found that strontium could partially substitute for Ca^{2+} but that magnesium and barium could not (Fig. 1H). This is reminiscent of many other Ca^{2+} binding proteins, which commonly have the capacity to bind Sr^{2+} due to their similar chemical properties {Lepsik 2007; Sandier 1999}. Selectivity against other alkaline-earth metals is achieved by the chemistry between the ion and binding site, e.g. nature of coordinating residues, bond lengths, coordination geometry, and level of ion-hydration (reviewed in {Gouaux 2005}).”

5) Although the new protein structure is apparently closed from the outside, the authors seem to suggest that the calcium enters from the outside environment, where the calcium concentration is high (around 1 mM); this is consistent with the binding constant determined. Looking at the structure, the calcium is quite deep into the protein and seems to be located closer to the inside environment. Could it possibly have entered from that side? I agree, it would seem unlikely given the known binding constant and the low calcium concentrations inside the cell. I guess the point of how the bound calcium may get to its position, should be more clearly addressed in the paper, to make it clearer for the reader.

As the reviewer mention, it is highly unlikely that the Ca binds from the inside where the Ca concentration is exceptionally low. In addition, removal of calcium from the outside of oocytes eliminates Fpn-mediated iron efflux (see Fig. 1). To clarify how the Ca ion gets to the site, we have amended the discussion of the transport model:

“Our results provide a framework for a transport model for the Fpn proteins (Fig. 4e-g). In the outward facing structure the Ca^{2+} site is solvent accessible. From this state, we propose that the binding of Ca^{2+} from extracellular fluid activates Fpn by triggering a conformational change that enable the transition from the open outward to open inward states.”

6) The ITC binding data seem of good quality and show endothermic binding of calcium. In the case of typical EF-hand calcium binding proteins the binding of calcium is usually exothermic. Clearly the calcium-binding site here must be quite different from the typical EF-hand sites. Having said that, the proposed coordination as a tetrahedral pyramid seems odd. Most high-resolution protein calcium-binding sites published to date seem to display 7-coordination in a pentagonal bipyramid arrangement, hence it seems like something is missing here. Are there maybe coordinating water molecules missing here that could further fill out the binding site? Perhaps the resolution in this part of the structure is not good enough to draw detailed conclusions about the calcium coordination?

We agree with the reviewer in that there are likely additional water molecules at the Ca^{2+} site, which are not resolved at the current resolution. To obtain a more complete crystallographic view of the Ca^{2+} coordination, a structure sub- 2.5Å would likely be required (Zheng *et al.*, *J Inorg Chem* 2008, PMID 18614239; Carugo and Bordo, *Acta Crystallogr D Biol Crystallogr* 1999, PMID 10089359). However, due to the small size of the crystals, we have not been able to improve beyond the current resolution. Future efforts will hopefully resolve further atomic details, although we would not anticipate this to change the main aspects of the binding site as presented in the manuscript. We trust the current statement in the manuscript leaves room for reinterpretation of the absolute coordination sphere.

“At the present resolution, the overall coordination sphere resembles that of a tetrahedral pyramid; however, it is possible that there could be unresolved water molecules that complete the coordination sphere, as in other structures^{14,15}.”

7) If I understand this correctly, the new structure was determined at pH 8, while previous structures of the same protein were done at pH5. If this is correct, this should be mentioned directly in the text in the results section, to make it easier for the reader to follow the subsequent discussion on the roles of histidine, etc.

The previously determined inward facing structure was crystallized at pH 5 (condition pH determined post crystallization to make sure), whilst the previous outward facing structure and new inward facing structures were crystallized at pH 8. As the reviewer suggests, readers will benefit from having this clarified. In response to this we have amended the second paragraph on pg 6.

“The previous structures of BbFpn were crystallised at pH 8 and pH 5 for the outward and inward facing structures, respectively. The low pH at which the inward facing structure was crystallised may unnaturally promote that state by having weakened salt bridges and hydrogen bonds that would otherwise stabilise the outward facing state⁸. In the structure presented here, crystallised at pH 8, the presence of Ca²⁺ appears to be catalysing the conformational change to the inward conformation.”

8) Page 4. Why was zinc transport not tested, given that the authors mention that it is a known ligand for transport? Likewise why was nickel transport tested here, even though it is apparently not a known ligand (at least it was not mentioned by the authors?)

We now demonstrate that human Fpn requires extracellular calcium for the transport of iron, cobalt, nickel, and zinc. We tested nickel in human Fpn (i) since our prior unpublished data revealed that human Fpn transports nickels, and (ii) since the BbFpn structure shown in Fig 6 contained Ni. We wished to verify that Ni was also a substrate of human Fpn and rule out the possibility that its appearance was not an artefact resulting from the crystallization conditions used.

9) I am confused about the appearance of the nickel-EDTA complex in the structure. His-tagged proteins are usually washed of the nickel-NTA columns with high concentrations of imidazole. I see in the methods that a TEV cleavage step was used and this does indeed require EDTA, but this apparently was followed by dialysis and then repurification over nickel-NTA. Hence it would seem that the EDTA was removed and from this procedure it seems more likely that nickel imidazole would be found bound to the protein. This raises a number of questions, is the EDTA density conclusive, or could it have been an imidazole complex instead? Does the protein crystallize in the same manner with its tag attached? If that were the case TEV and EDTA would not be needed during the protein preparation and there likely should be no bound nickel-EDTA complex found. Finally have such ‘contaminating’ nickel-EDTA complexes been observed before in other published protein structures; if so this should be commented on in the supplementary material (given that His-tags and such nickel columns are in extremely widespread use). If not commonly seen, is this an anomalous observation and why do the authors feel that it happens here?

We initially shared the reviewers’ surprise regarding the bound Ni-EDTA complex. However, the electron density for the EDTA molecule is unambiguous, and using ITC we measured the BbFpn affinity for Ni-EDTA to be high.

Following the first elution of the BbFpn-TEV-GFP-His, the protein is mixed with His-TEV (with EDTA) in an overnight dialysis. In the following step, the mixture is poured over Ni-NTA, at which stage His-TEV and GFP-His rebinds, whilst BbFpn and other constituents (including EDTA) are collected in the flow through. Hence, we believe that the Ni-EDTA complex binds tightly enough to follow through the remaining purification step. We have amended the Methods section to clarify this.

“The GFP was cleaved from the eluted protein using His-tagged TEV protease (containing 0.5 mM EDTA) under dialysis with buffer containing 20 mM Tris pH 8.0 and 300 mM NaCl (overnight). The GFP moiety and TEV protease were cleared from BbFpn by re-binding the dialysed protein to Ni-NTA (i.e. re-binding of His-tagged GFP and TEV). The untagged BbFpn protein (including EDTA) was eluted and concentrated to 0.5 mL before loading onto a Superdex 200 16/600 size exclusion column (GE Healthcare Life Sciences), pre-equilibrated with 20 mM Tris pH 8.0, 300 mM NaCl, and 0.004% LMNG.”

That mutating, in human Fpn, the residue corresponding to the Ni binding site in BbFpn changes the metal specificity of human Fpn (Fe > Ni) to a Ni-preferring transporter provides strong evidence for specific binding of Ni in the Fpn family and argues against the presence of nickel in the BbFpn structure being ‘anomalous’.

Interestingly, the finding of an EDTA molecule at the substrate-binding site of a metal binding protein is not unprecedented. The Ni-binding protein NikA has previously been found to bind Ni-EDTA (and other metabolites; Cherrier *et al.*, *JACS* 2005: <https://www.ncbi.nlm.nih.gov/pubmed/16011372>), whilst the physiological substrate is believed to be Ni in complex with an endogenous metal chelator (Cherrier *et al.*, *Biochemistry* 2008: <https://www.ncbi.nlm.nih.gov/pubmed/18759453>). The overall architecture in the binding sites of BbFpn and NikA share some similarities – a conserved Arg residue forming a bidentate ligand to one of the carboxylate groups of the EDTA molecule, and with the metal coordinated by a His residue. The full relevance of this for the Fpn proteins is not yet clear, but ongoing and future work is aimed at determining this and identifying any putative co-substrate for the Fpn transporters.

REVIEWERS' COMMENTS:

Reviewer #2 (Remarks to the Author):

The Authors made efforts to improve the manuscript according to this reviewer's suggestions. Although I am somewhat disappointed by the fact that they did not succeed in running iron transport assays in bacteria, I would now recommend publication.

Reviewer #3 (Remarks to the Author):

The authors addressed most of the issues raised by the reviewers in an appropriate way.